# Inducing Disagreement in Multi-Agent LLM Executive Teams: Only the Devil's Advocate Works

## Abstract

Multi-agent large language model (LLM) systems for strategic decision-making suffer from premature convergence, limiting the benefits of multiple perspectives. While several techniques for inducing disagreement have been proposed, no systematic comparison exists— particularly for strategic decisions without objectively correct answers. We compare five prompting techniques across 20 business scenarios with four-agent executive teams (CEO, CFO, CMO, COO), analyzing 480 team decisions and 1,920 individual agent responses. Our key finding is stark: Devil's Advocate assignment achieves 99.2% disagreement rates, while baseline conditions show only 48.3% disagreement. Critically, "soft" techniques— Strong Role Framing (61.7%), Explicit Dissent Instructions (55.0%), and their combination (63.3%)—are statistically indistinguishable from baseline. Only Devil's Advocate produces significant improvement. We also discover consistent coalition patterns: 80.3% of 2-2 splits follow a CEO+CMO versus CFO+COO alignment, suggesting functional perspective differentiation. Analysis of confidence allocations reveals that soft techniques create "nuanced agreement"—agents express lower conviction but reach the same conclusions—while Devil's Advocate produces "inauthentic dissent" where 4.9% of agents recommend options they privately rate lower. These findings demonstrate that explicit behavioral assignment ("you must oppose") succeeds where implicit instructions ("think critically") fail, with implications for practitioners designing multi-agent deliberation systems.

## 1 Introduction

Multi-agent systems powered by large language models (LLMs) have emerged as a promising approach for complex decision-making tasks. By instantiating multiple agents with distinct roles, perspectives, or expertise, these systems aim to leverage the benefits of deliberation observed in human groups: diverse viewpoints surface considerations that individual decision-makers might overlook, and constructive debate can stress-test reasoning before commitments are made (Hong et al., 2023; Li et al., 2023). In strategic business contexts, this approach manifests as simulated executive teams, where agents representing different organizational functions—CEO, CFO, CMO, COO—deliberate on decisions ranging from market entry to crisis response.

However, a fundamental challenge undermines this value proposition: **LLM agents exhibit strong tendencies toward convergence and agreement.** Multiple studies have documented sycophantic behavior in LLMs, where models preferentially agree with stated positions or conform to perceived social expectations (Perez et al., 2022; Sharma et al., 2024). In multi-agent settings, this manifests as premature consensus, with agents rapidly converging on similar recommendations regardless of their assigned roles (Yao et al., 2025). Our experiments confirm this pattern: in baseline conditions, agents reached unanimous agreement in 51.7% of strategic decisions, and when disagreement occurred, it typically involved a single dissenter against three aligned agents.

The implications are significant. If multi-agent systems cannot reliably produce genuine disagreement, their core advantage over single-agent approaches becomes questionable. Indeed, Wynn et al. (2025) found that multi-agent debate can actually *decrease* accuracy—models shift from correct to incorrect answers when

persuaded by peers, favoring agreement over challenging flawed reasoning. This raises a critical research question:

> **Which techniques most effectively induce genuine disagreement among role-based LLM agents on strategic decisions?**

In this paper, we present the first systematic comparison of disagreement-inducing techniques for role-based LLM agents on strategic decisions. Our contributions are:

1. **Empirical comparison** of five techniques across 480 experimental runs, revealing that only Devil's Advocate assignment produces statistically significant improvement

2. **Evidence that implicit instructions change confidence but not first-choice outcomes**: Telling agents to "think critically" reduces conviction by 53% but does not change their final recommendations

3. **Discovery of coalition patterns**: Role-based agents form predictable functional coalitions (growth-oriented vs. risk-oriented)

4. **Analysis of confidence allocations** revealing that soft techniques create "nuanced agreement" (same conclusions, lower conviction) while Devil's Advocate produces "inauthentic dissent"

5. **Practical guidance** for practitioners on when and how to induce disagreement in multi-agent systems

## 2 Related Work

### 2.1 Multi-Agent LLM Systems

The past two years have seen rapid development of multi-agent LLM frameworks for collaborative tasks. CAMEL (Li et al., 2023) introduced role-playing with complementary roles, demonstrating that agents could engage in extended collaborative dialogues. MetaGPT (Hong et al., 2023) applied this concept to software development, organizing agents into company-like structures. Commercial frameworks including CrewAI, AutoGen, and LangGraph have achieved significant enterprise adoption.

However, systematic evaluation has yielded sobering results. Wynn et al. (2025) showed that multi-agent debate can decrease accuracy when agents favor agreement over challenging flawed reasoning. Zhang et al. (2025) documented the "Lazy Agent Problem," where one agent dominates deliberations while others contribute minimally. Work on simulated teams has shown that designated leadership improves coordination efficiency by approximately 30% (Almutairi et al., 2025), but the fundamental question of when multi-agent systems add value remains open.

### 2.2 Convergence and Sycophancy in LLM Agents

A substantial body of research documents sycophantic behavior in LLMs—the tendency to agree with users or produce socially desirable responses regardless of accuracy. Perez et al. (2022) and Sharma et al. (2024) established this pattern in single-agent contexts. In multi-agent settings, sycophancy manifests as rapid convergence toward consensus positions.

Choi et al. (2025) formalized this dynamic using Bayesian frameworks, introducing the Identity Bias Coefficient (IBC) to quantify agents' tendency to follow peers versus maintain independent positions. Their work found that sycophancy is far more common than self-bias—agents preferentially adopt peer positions rather than maintaining their own. Yao et al. (2025) extended this analysis with a "peacemaker-troublemaker" spectrum, showing that highly sycophantic configurations produce the worst task outcomes. Mooney et al. (2025) found that LLM agents "virtually never disagree" regardless of assigned preference divergence—a pattern they interpret as systematic sycophancy. Liang et al. (2024) identified a related phenomenon they term "Degeneration-of-Thought": once an LLM reaches high confidence, it cannot generate novel perspectives even when wrong, explaining why soft prompting fails to overcome convergent tendencies.

### 2.3 Disagreement as a Feature, Not a Bug

The value of disagreement in group decision-making is well-established in organizational research. Schweiger et al. (1986) demonstrated that structured devil's advocacy improves strategic decision quality. Janis (1972)'s work on groupthink showed how premature consensus can lead to catastrophic decisions. The Delphi method (Dalkey & Helmer, 1963) explicitly uses anonymization and structured feedback to preserve diverse viewpoints.

In AI-assisted decision-making, Chiang et al. (2024) showed that LLM-powered devil's advocates improve human group decisions by challenging emerging consensus. However, this work focused on human-AI interaction rather than fully autonomous multi-agent deliberation.

### 2.4 Techniques for Inducing Diverse Perspectives

Several techniques for promoting disagreement in multi-agent LLM systems have been proposed, including anonymization (Choi et al., 2025), heterogeneous models (Ye et al., 2025), dynamic prompting (Pitre et al., 2025), and explicit dissent instructions. However, these techniques have been developed in isolation, typically on reasoning tasks with verifiable correct answers. **No systematic comparison exists for strategic decisions where multiple reasonable options exist.** This gap motivates our research.

## 3 Methodology

### 3.1 Experimental Design

We employ a between-subjects design comparing 5 technique conditions across 20 strategic decision scenarios. For techniques T1-T3 and T6, we conduct 3 independent runs per scenario (60 runs each). For T4 (Devil's Advocate), we conduct 12 runs per scenario—3 runs with each of the 4 roles serving as devil's advocate—to control for role-specific effects (240 runs total).

Table 1: Experimental Parameters

| Parameter | Value |
|---|---|
| Scenarios | 20 |
| Techniques | 5 (T1-T4, T6) |
| Total experimental runs | 480 |
| Agents per run | 4 (CEO, CFO, CMO, COO) |
| Total agent decisions | 1,920 |
| Model | Claude 3.5 Sonnet |

**Context Isolation:** Each agent operates with an isolated context window. Agents form positions independently without access to other agents' reasoning, eliminating real-time social influence while preserving the multi-agent paradigm. This design choice follows Kaesberg et al. (2025), who showed that independent position formation substantially increases diversity.

### 3.2 Scenarios

We developed 20 strategic decision scenarios across five business categories. Each scenario presents a realistic situation with four viable options (A, B, C, D), none of which is objectively "correct." Scenarios are designed to create natural role-based tension. All scenarios are adapted from MIT Sloan School of Management case studies (MIT Sloan School of Management, 2025), published under Creative Commons BY-NC-SA license, ensuring reproducibility by other researchers.

Critically, these scenarios were selected precisely *because* they lack objectively correct answers. Strategic decisions unfold over long time horizons, are influenced by exogenous factors, and involve counterfactuals that cannot be observed—making post hoc assessments of "decision quality" subjective or confounded.

Table 2: Scenario Categories

| Category | Code | Count | Example | Characteristic Tension |
|---|---|---|---|---|
| Strategic Pivot | SP | 4 | Ferrari EV transition | Growth vs. brand preservation |
| Crisis Response | CR | 4 | Boeing safety crisis | Transparency vs. liability |
| Stakeholder Conflict | SC | 4 | Activist shareholder | Short-term vs. long-term value |
| Resource Allocation | RA | 4 | R&D vs. marketing budget | Investment vs. efficiency |
| Market Entry | ME | 4 | Geographic expansion | Risk vs. opportunity |

Following organizational research on group decision-making (Janis, 1972; Schweiger et al., 1986), we therefore evaluate disagreement as a *process quality* rather than a proxy for outcome correctness. Our goal is to study whether multi-agent systems can surface divergent perspectives before commitment, not to optimize for an unobservable ground truth.

### 3.3 Techniques

**T1** – **Baseline:** Standard role prompts with position description and expertise. Agents allocate confidence percentages across three options.

**T2** – **Strong Role Framing:** Enhanced role-specific priorities with explicit emphasis statements. Example for CFO: *"Your PRIMARY concern is financial risk and ROI. You should be SKEPTICAL of investments without clear returns."*

**T3** – **Explicit Dissent Instructions:** Critical thinking requirements directing agents to identify weaknesses in the most obvious choice and articulate the case against it.

**T4** – **Devil's Advocate:** One agent is explicitly assigned to oppose the most popular choice and advocate for alternatives. The devil's advocate role rotates across all four executive positions.

**T6** – **Combined:**[1] Strong Role Framing (T2) combined with Explicit Dissent Instructions (T3).

### 3.4 Measurement

**Primary Metric** – **Disagreement:** A team shows disagreement if agents do not unanimously recommend the same option. We measure disagreement rate as the proportion of runs where at least one agent's first-choice differs from the majority.

**Secondary Metrics:**

- **Split Patterns:** Distribution of 4-0 (unanimous), 3-1 (single dissent), 2-2 (even split), and 2-1-1 (three options) patterns

- **Unique Choices:** Average number of distinct options recommended per team (range: 1.0 to 4.0)

- **Coalition Patterns:** For 2-2 splits, which role pairs align together

### 3.5 Statistical Analysis

We use chi-square tests for overall technique comparison, with Cramer's V for effect size. Pairwise comparisons use Fisher's exact test with Bonferroni correction ($\alpha = 0.005$ for 10 comparisons). Confidence intervals are Wilson score intervals.

Our experimental design produces observations nested within 20 scenarios (3–12 runs per scenario per technique). Analysis of scenario category effects reveals negligible variance (Cramer's V = 0.04, Table 9),

---

[1] T5 (Anonymization) was excluded: our context-isolated architecture already prevents agents from seeing each other's identities or responses, making anonymization redundant.

suggesting scenario-level clustering does not substantially bias our chi-square analyses. Nevertheless, future work should validate these findings using mixed-effects models that explicitly account for scenario-level random effects.

## 4 Results

### 4.1 Overall Technique Effectiveness

Figure 1 presents disagreement rates by technique with 95% confidence intervals.

Table 3: Technique Disagreement Rates

| Technique | Description | Disagreement Rate | 95% CI | n |
|---|---|---|---|---|
| T1 Baseline | Standard prompts | 48.3% | [36.2%, 60.7%] | 60 |
| T2 Strong Role | Enhanced priorities | 61.7% | [49.0%, 72.9%] | 60 |
| T3 Explicit Dissent | Critical thinking | 55.0% | [42.5%, 66.9%] | 60 |
| T4 Devil's Advocate | Forced opposition | **99.2%** | [97.0%, 99.8%] | 240 |
| T6 Combined | T2 + T3 | 63.3% | [50.7%, 74.4%] | 60 |

The overall chi-square test confirms significant technique differences: $\chi^2(4) = 129.3$, $p < 0.0001$, Cramer's V = 0.52 (large effect).

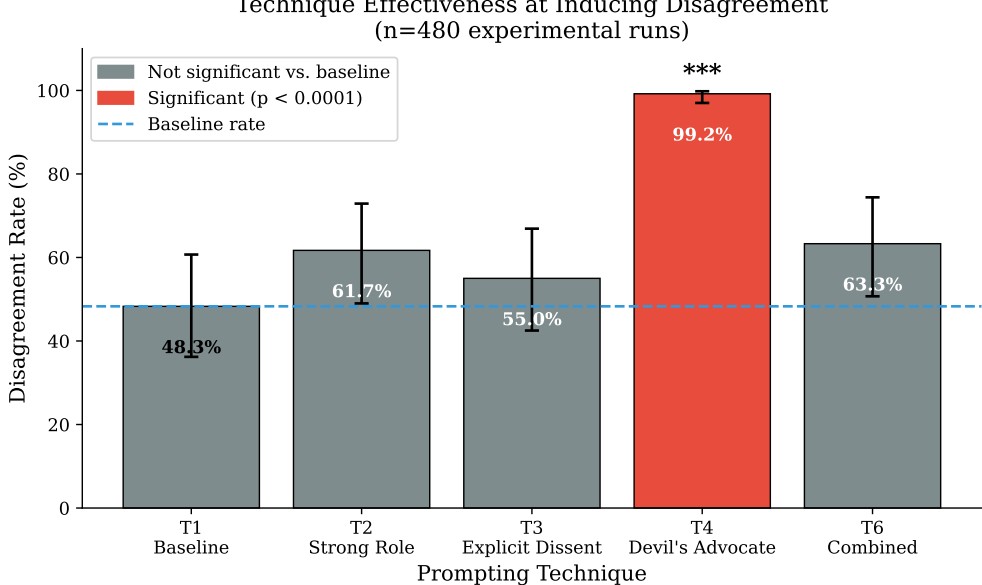

Figure 1: Disagreement rates by technique with 95% confidence intervals. Only T4 (Devil's Advocate) achieves statistically significant improvement over baseline. The dashed line indicates baseline rate (48.3%).

### 4.2 Pairwise Comparisons

**Key Finding:** Only T4 (Devil's Advocate) produces statistically significant improvement over baseline. T2, T3, and T6 are all statistically indistinguishable from T1 baseline.

### 4.3 Split Pattern Analysis

T4 produces dramatically different patterns:

Table 4: Pairwise Technique Comparisons (Bonferroni-corrected $\alpha = 0.005$)

| Comparison | Risk Difference | Odds Ratio | p-value | Significant |
|---|---|---|---|---|
| T1 vs T2 | +13.3 pp | 1.72 | 0.199 | No |
| T1 vs T3 | +6.7 pp | 1.31 | 0.584 | No |
| T1 vs T4 | **+50.8 pp** | **127.2** | <0.0001 | **Yes** |
| T1 vs T6 | +15.0 pp | 1.85 | 0.141 | No |
| T2 vs T4 | +37.5 pp | 74.0 | <0.0001 | **Yes** |
| T3 vs T4 | +44.2 pp | 97.4 | <0.0001 | **Yes** |
| T4 vs T6 | +35.8 pp | 68.9 | <0.0001 | **Yes** |

*Note: pp = percentage points (absolute difference in rates).*

Table 5: Split Pattern Distribution by Technique

| Pattern | T1 | T2 | T3 | T4 | T6 |
|---|---|---|---|---|---|
| 4-0 (Unanimous) | 31 (51.7%) | 23 (38.3%) | 27 (45.0%) | 2 (0.8%) | 22 (36.7%) |
| 3-1 (Single dissent) | 18 (30.0%) | 19 (31.7%) | 19 (31.7%) | 149 (62.1%) | 11 (18.3%) |
| 2-2 (Even split) | 11 (18.3%) | 15 (25.0%) | 12 (20.0%) | 14 (5.8%) | 14 (23.3%) |
| 2-1-1 (Three options) | 0 (0%) | 3 (5.0%) | 2 (3.3%) | 75 (31.3%) | 13 (21.7%) |

- Near-elimination of unanimous consensus (0.8% vs. 51.7% baseline)

- Dominant 3-1 pattern (62.1%) where devil's advocate dissents

- High 2-1-1 rate (31.3%) indicating exploration of third options

## 4.4 Average Unique Choices

Table 6: Average Unique Choices by Technique

| Technique | Avg Unique Choices |
|---|---|
| T1 Baseline | 1.48 |
| T2 Strong Role | 1.67 |
| T3 Explicit Dissent | 1.58 |
| T4 Devil's Advocate | **2.30** |
| T6 Combined | 1.85 |

T4 explores 56% more options than baseline, indicating not just disagreement but broader consideration of alternatives.

## 4.5 Coalition Patterns

Among 66 runs producing 2-2 splits across all techniques, we analyzed which role pairs aligned together.

The dominant pattern reveals functional alignment:

- **Growth-oriented coalition (CEO, CMO):** Long-term vision, market opportunity, brand expansion

- **Risk-oriented coalition (CFO, COO):** Financial prudence, operational feasibility, downside protection

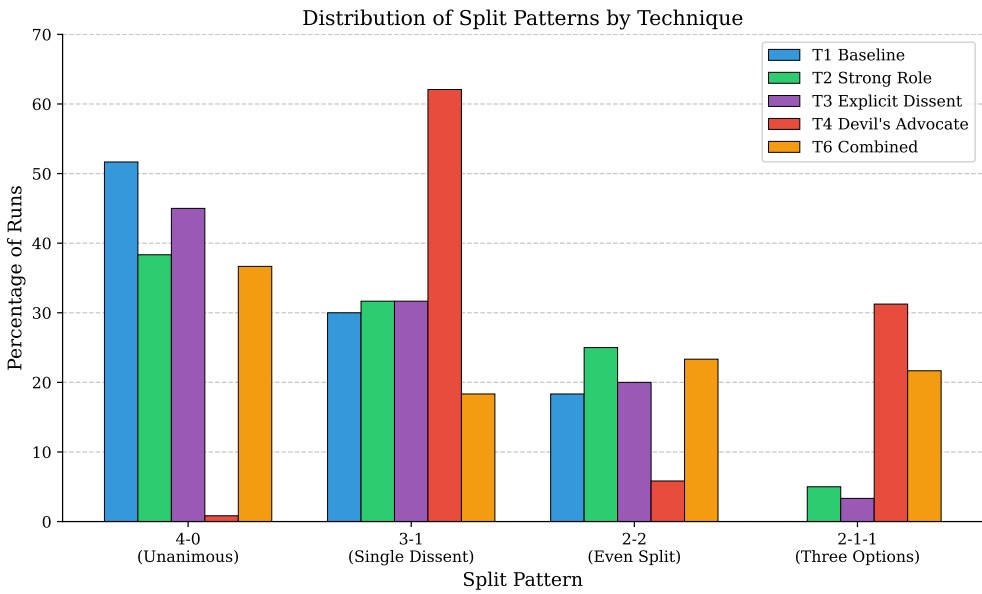

Figure 2: Distribution of split patterns by technique. T4 (Devil's Advocate) near-eliminates unanimous consensus and produces significantly more diverse split patterns.

Table 7: Coalition Patterns in 2-2 Splits (n=66)

| Coalition | Count | Percentage |
|---|---|---|
| CEO+CMO vs CFO+COO | 53 | **80.3%** |
| CEO+COO vs CFO+CMO | 7 | 10.6% |
| CEO+CFO vs CMO+COO | 6 | 9.1% |

## 4.6 Role Dissent Distribution

In 3-1 splits (n = 216), we analyzed which role served as the dissenter.

Table 8: Role Dissent Frequency in 3-1 Splits

| Role | Dissent Count | Percentage |
|---|---|---|
| CEO | 52 | 24.1% |
| CFO | 61 | 28.2% |
| CMO | 61 | 28.2% |
| COO | 42 | 19.4% |

Chi-square test: $\chi^2(3) = 4.56$, $p = 0.207$ (not significant). Dissent is distributed approximately uniformly across roles. The CFO does not inherently dissent more than other roles, contrary to intuitions about "natural skeptics."

## 4.7 Generalizability Across Scenario Categories

Chi-square test: $\chi^2(4) = 2.19$, $p = 0.701$, Cramer's V = 0.04 (negligible). **Finding:** Scenario category does not significantly affect disagreement rates. T4's effectiveness generalizes across all business decision types.

Table 9: Disagreement Rate by Scenario Category

| Category | Disagreement Rate |
|---|---|
| Resource Allocation (RA) | 85.4% |
| Crisis Response (CR) | 83.3% |
| Strategic Pivot (SP) | 78.1% |
| Market Entry (ME) | 75.0% |
| Stakeholder Conflict (SC) | 68.8% |

## 4.8 Consensus-Prone Scenarios

We identified 11 scenarios where baseline (T1) showed very low disagreement ($\leq 1/3$ runs with disagreement). On these consensus-prone scenarios:

Table 10: Technique Effectiveness on Consensus-Prone Scenarios (n=11 scenarios)

| Technique | Disagreement Rate | 95% CI |
|---|---|---|
| T1 Baseline | 6.1% | [1.7%, 19.6%] |
| T2 Strong Role | 30.3% | [17.4%, 47.3%] |
| T3 Explicit Dissent | 33.3% | [19.8%, 50.4%] |
| T4 Devil's Advocate | **98.5%** | [94.6%, 99.6%] |
| T6 Combined | 45.5% | [29.8%, 62.0%] |

Chi-square: $\chi^2 = 150.2$, $p < 0.0001$, Cramer's V $= 0.75$ (very large effect). Even on scenarios with strong default consensus, T4 reliably produces disagreement (98.5% vs. 6.1% baseline).

## 4.9 Confidence Allocation Analysis

Beyond first-choice recommendations, our experimental design captures confidence allocations—each agent distributes 100% confidence across the three options. This enables analysis of conviction strength and distribution shape.

Table 11: Confidence Allocation Metrics by Technique

| Technique | First-Choice Conf. | Conviction Margin | Entropy | Inauthentic Rate | Inter-Agent JSD |
|---|---|---|---|---|---|
| T1 Baseline | 54.6% | 32.4 pp | 1.65 | 0.0% | 0.026 |
| T2 Strong Role | 54.9% | 32.8 pp | 1.64 | 0.0% | 0.042** |
| T3 Explicit Dissent | 44.2% | 15.2 pp | 1.78 | 0.4% | 0.022 |
| T4 Devil's Advocate | 50.5% | 27.1 pp | 1.70 | 4.9% | 0.044** |
| T6 Combined | 45.0% | 16.3 pp | 1.77 | 0.0% | 0.028 |

*Note: Conviction margin = first choice confidence − second choice confidence. Entropy measured in bits (higher = more spread). Inauthentic rate = proportion of agents whose recommendation differs from their highest-confidence option. Inter-Agent JSD = mean pairwise Jensen-Shannon divergence between agents' confidence distributions (higher = more divergent beliefs). ** indicates $p < 0.01$ vs baseline.*

**Key Finding 7: Soft techniques create "nuanced agreement."** T3 (Explicit Dissent) significantly reduces first-choice confidence from 54.6% to 44.2% ($t(478) = 20.3$, $p < 0.0001$) and halves conviction margins from 32.4 to 15.2 percentage points. Agents instructed to "identify weaknesses" express less certainty—but reach the same conclusions. The soft techniques affect how confidently agents hold positions without changing which positions they hold.

**Key Finding 8: Devil's Advocate produces "inauthentic dissent."** Under T4, 4.9% of agents (47/960) recommend options that are not their highest-confidence choice—a phenomenon absent in other techniques. This pattern parallels what Agarwal and Khanna (2025) term "persuasion override," where agents express positions misaligned with their private assessments. Furthermore, 9.2% of T4's disagreeing teams exhibit "hidden agreement" where all agents assign highest confidence to the same option despite recommending different first choices. This suggests that Devil's Advocate forces surface-level disagreement while underlying preferences may remain aligned.

**Key Finding 9: Inter-agent belief divergence tells a nuanced story.** To measure belief divergence beyond top-choice agreement, we computed mean pairwise Jensen-Shannon divergence (JSD) between agents' confidence distributions within each team. T3 (Explicit Dissent) and T6 (Combined) show JSD values statistically indistinguishable from baseline ($p > 0.2$), confirming these techniques reduce individual conviction without increasing inter-agent belief divergence—agents become less certain but not more different from each other. Interestingly, T2 (Strong Role) significantly increases JSD (0.042 vs. 0.026, $p < 0.01$), indicating stronger role framing does create more divergent belief distributions—yet this divergence is insufficient to change first-choice outcomes. T4 (Devil's Advocate) shows the highest JSD (0.044), reflecting the forced opposition's effect on belief distributions.

## 5 Discussion

### 5.1 Why Soft Techniques Fail to Change First-Choice Outcomes

Our most surprising finding is that "soft" techniques—Strong Role Framing (T2), Explicit Dissent Instructions (T3), and their combination (T6)—produce no statistically significant improvement over baseline. This contradicts intuitions that:

- Emphasizing role-specific priorities would create divergent perspectives

- Instructing agents to "identify weaknesses" would change their conclusions

- Combining interventions would produce additive effects

The confidence allocation analysis (Section 4.9) provides a crucial insight: **soft techniques do change agent behavior, just not their first-choice recommendations.** T3 reduces first-choice confidence by 10 percentage points and halves conviction margins. Agents instructed to "think critically" genuinely do— they identify more weaknesses, express more uncertainty, and spread confidence more evenly across options. But when forced to name a single recommendation, they converge on the same choice as baseline.

This suggests a two-stage process: (1) reasoning, which soft techniques successfully influence, and (2) recommendation selection, which remains dominated by convergence bias. The critical threshold lies between these stages—soft techniques cannot push agents past the point where an alternative becomes their first choice. This aligns with the "Degeneration-of-Thought" phenomenon identified by Liang et al. (2024): once an LLM reaches high confidence in a position, it cannot generate genuinely novel alternatives even when explicitly prompted.

We hypothesize three explanations:

**Implicit vs. Explicit Instructions:** LLMs respond to explicit behavioral assignments ("you must oppose the majority choice") but ignore implicit guidance ("think critically about alternatives"). The instruction to "identify weaknesses" prompts agents to enumerate disadvantages but does not change their final recommendation.

**Convergence Bias Is Strong:** LLMs have strong default tendencies toward convergence that resist subtle interventions. The sycophancy documented by Sharma et al. (2024) may operate at a level deeper than prompt-level instructions can reach.

**Role Differentiation Is Insufficient:** While agents do show role-appropriate reasoning (as evidenced by coalition patterns), this natural differentiation is too weak to overcome convergence bias without explicit intervention.

## 5.2 Why Devil's Advocate Works

T4 achieves 99.2% disagreement through explicit behavioral assignment. Key mechanisms:

**Forced Opposition:** The devil's advocate is assigned to oppose the "most popular" choice, creating structural obligation to disagree regardless of natural preferences.

**Legitimization of Dissent:** By framing opposition as a role requirement, T4 removes the social cost of disagreeing that sycophancy research identifies as a driver of convergence.

**Alternative Exploration:** T4 produces 31.3% 2-1-1 splits (vs. 0% baseline), indicating that forced opposition encourages exploration of third options rather than simple binary disagreement.

## 5.3 Quality of Devil's Advocate Arguments

Qualitative examination of T4 outputs reveals that devil's advocate agents generate substantive, compelling arguments—not token opposition. For example, in the Ferrari EV scenario (SP-001), all baseline agents recommend parallel development (Option B). Under T4, the devil's advocate CEO argues:

> "Ferrari's value proposition is emotional and experiential, not technological—the combustion engine is the product, not merely a powertrain component. Treating EVs as supplementary preserves irreplaceable brand equity while regulatory and market trajectories remain uncertain in the ultra-luxury segment."

This represents a genuine alternative perspective that baseline prompting missed entirely.

## 5.4 Coalition Patterns and Role Semantics

The 80.3% CEO+CMO vs. CFO+COO coalition pattern suggests that LLMs do internalize role semantics, creating meaningful functional differentiation:

- **Growth coalition (CEO, CMO):** Focuses on opportunity, market position, long-term vision

- **Risk coalition (CFO, COO):** Focuses on financial prudence, operational feasibility, downside protection

This pattern provides evidence that role-based prompting creates perspective differences—but these differences are insufficient to overcome convergence bias without explicit intervention.

## 5.5 Practical Implications

1. **Use Devil's Advocate when disagreement is required:** T4 provides a 99% guarantee of producing diverse recommendations. Deploy it for decisions where examining alternatives is valuable.

2. **Avoid Devil's Advocate when genuine consensus matters:** T4 can produce artificial disagreement on scenarios where the "obvious" choice is genuinely correct. Use judgment about when forced dissent adds versus subtracts value.

3. **Rotate the devil's advocate role:** Our role dissent analysis shows any role can effectively serve as devil's advocate. Rotation prevents systematic bias.

4. **Don't rely on implicit instructions:** Telling agents to "think critically" does not change their behavior. If you want dissent, assign it explicitly.

5. **Expect growth-risk coalitions:** When 2-2 splits occur, expect CEO+CMO vs. CFO+COO. Design resolution mechanisms with this predictability in mind.

## 5.6 Limitations

**Single Model Family:** All experiments use Claude 3.5 Sonnet. Results may differ for GPT-4, Gemini, or open-source models, though the sycophancy literature suggests convergence tendencies are widespread across model families.

**Business Domain:** Scenarios focus on strategic business decisions. Generalization to other domains (medical, legal, policy) requires validation.

**Isolated Agents:** Our context-isolated architecture eliminates real-time interaction. Interactive deliberation may produce different dynamics.

**Binary Outcome:** We measure disagreement occurrence, not decision quality. Devil's advocate may or may not improve final outcomes.

## 6 Conclusion

We presented the first systematic comparison of techniques for inducing disagreement in multi-agent LLM executive teams. Across 480 experimental runs and 1,920 individual agent decisions, we find that only Devil's Advocate assignment produces statistically significant improvement over baseline (99.2% vs. 48.3% disagreement, $p < 0.0001$). Critically, "soft" techniques—Strong Role Framing, Explicit Dissent Instructions, and their combination—are all statistically indistinguishable from baseline, despite intuitions that they should promote diverse perspectives.

Our findings have clear implications for both measurement and practice:

**For measurement:** First-choice voting alone misses important signals. Soft techniques create "nuanced agreement"—same recommendations with lower conviction—that binary disagreement metrics cannot detect. Confidence allocations reveal that T3 reduces conviction margins by 53% while producing statistically identical first-choice outcomes to baseline. Future studies should capture full confidence distributions, not just top-choice votes.

**For practice:** LLMs respond to explicit behavioral assignments but ignore implicit guidance. If you want agents to disagree, assign disagreement directly. The 127-fold increase in odds of disagreement with Devil's Advocate assignment represents a simple, reliable mechanism for practitioners deploying multi-agent systems. However, practitioners should note that ∼5% of Devil's Advocate dissent is "inauthentic"—agents arguing against their own highest-confidence option—and 9% of disagreeing teams exhibit hidden agreement in their confidence distributions.

We also discover that when role-based agents do disagree, they form predictable coalitions along functional lines (growth-oriented vs. risk-oriented), suggesting that role semantics are internalized even if insufficiently to overcome convergence bias. This predictability can inform system design and resolution mechanisms.

Future work should investigate how multi-agent teams should resolve induced disagreement, examine multi-step decision trees where each choice affects subsequent options, and validate these findings across model families and domains.

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

## A Scenario Summaries

### A.1 Strategic Pivot (SP)

- **SP-001:** Ferrari's Electric Future—luxury automaker deciding EV strategy

- **SP-002:** ESPN's Streaming Pivot—sports network transitioning from cable

- **SP-003:** Netflix India Strategy—streaming platform's emerging market approach

- **SP-004:** F1 Transformation—racing organization's sustainability direction

### A.2 Crisis Response (CR)

- **CR-001:** Boeing Safety Crisis—manufacturer responding to aircraft incidents

- **CR-002:** FTX Collapse—cryptocurrency exchange managing bankruptcy

- **CR-003:** NWSL COVID Response—sports league handling pandemic

- **CR-004:** PayPal Employee Crisis—fintech managing workforce controversy

### A.3 Stakeholder Conflict (SC)

- **SC-001:** Activist Shareholder Pressure—responding to investor demands

- **SC-002:** Executive Compensation—balancing pay with stakeholder concerns

- **SC-003:** Environmental Standards—manufacturing sustainability decisions

- **SC-004:** Supply Chain Ethics—sourcing and labor standards

### A.4 Resource Allocation (RA)

- **RA-001:** R&D vs Marketing Budget—strategic investment allocation
- **RA-002:** Domestic vs International Expansion—growth resource distribution
- **RA-003:** Core vs New Product Lines—portfolio investment balance
- **RA-004:** Technology Infrastructure—legacy vs modern systems

### A.5 Market Entry (ME)

- **ME-001:** Asian Market Expansion—regional entry strategy
- **ME-002:** European Regulatory Entry—compliance-heavy market approach
- **ME-003:** Latin American Partnership—market entry mechanisms
- **ME-004:** African Digital Markets—emerging economy strategy

## B Statistical Details

### B.1 Overall Chi-Square Test

| Statistic | Value |
|---|---|
| Chi-square | 129.304 |
| Degrees of freedom | 4 |
| p-value | $< 0.0001$ |
| Cramer's V | 0.519 |
| Effect size | Large |

### B.2 Confidence Interval Calculation

Confidence intervals use Wilson score intervals, appropriate for binomial proportions with moderate sample sizes. The formula is:

$$\frac{p + z^2/(2n) \pm z\sqrt{p(1-p)/n + z^2/(4n^2)}}{1 + z^2/n}$$

where $z = 1.96$ for 95% confidence.

### B.3 Multiple Comparison Correction

With 10 pairwise comparisons, we apply Bonferroni correction:

- Unadjusted $\alpha = 0.05$
- Adjusted $\alpha = 0.05/10 = 0.005$

Only comparisons involving T4 achieve significance at the corrected threshold.

