# OpenReview forum: "Inducing Disagreement in Multi-Agent LLM Executive Teams: Only the Devil’s Advocate Works"
_TMLR — Rejected by TMLR_

### Review · Reviewer_yZ4c · 2026-02-25

**Summary Of Contributions:**

This paper presents a systematic empirical comparison of five prompting techniques designed to induce disagreement among role-based LLM agents in strategic decision-making contexts. Using a four-agent executive team setup (CEO, CFO, CMO, COO) across 20 business scenarios, the authors find that only the Devil's Advocate technique (T4) produces statistically significant improvement over baseline (99.2% vs. 48.3% disagreement rate), while "soft" techniques such as Strong Role Framing, Explicit Dissent Instructions, and their combination are statistically indistinguishable from baseline. Secondary findings include a consistent CEO+CMO vs. CFO+COO coalition pattern (80.3% of 2-2 splits) and evidence that soft techniques create "nuanced agreement" (reduced conviction without changed recommendations) rather than genuine divergence.

**Audience:**

Yes

**Audience Explanation:**

The problem of sycophancy and premature convergence in multi-agent LLM systems is a legitimate and practically important research question that will interest a portion of TMLR's audience. The finding that explicit behavioral assignment outperforms implicit prompting guidance is directionally useful for practitioners, even if the current evidence base is insufficient to make strong scientific claims. The confidence allocation analysis, distinguishing nuanced agreement from genuine divergence, also offers a methodological framing that could be valuable to the community if developed more rigorously.

**Broader Impact Concerns:**

No significant concerns beyond standard considerations for AI decision-support systems. The authors should briefly note that Devil's Advocate deployment in real-world decision systems could introduce artificial contrarianism in contexts where strong consensus reflects genuine correctness, potentially degrading decision quality. This risk is mentioned in passing in Section 5.5 but warrants a more prominent caveat.

**Claims And Evidence:**

No

**Claims Explanation:**

While the statistical analyses are conducted competently, the evidence does not convincingly support the paper's central claims for the following reasons:
1. The foundational premise is unjustified. The paper assumes throughout that higher disagreement rates are desirable without providing a principled justification for this in the LLM multi-agent setting. The cited organizational behavior literature (Janis, 1972; Schweiger et al., 1986) concerns human teams, and the applicability of these findings to LLM agents — which share the same underlying base model, training distribution, and inductive biases — is not established. Disagreement between instances of the same model may reflect sampling noise rather than genuine cognitive diversity, a distinction the paper never addresses.
2. The evaluation metric is critically insufficient. Disagreement rate as the sole primary metric cannot distinguish between productive divergence and noise. The paper's own findings undermine this metric: 4.9% of Devil's Advocate agents recommend options they privately rank lower ("inauthentic dissent"), and 9.2% of T4 disagreeing teams exhibit "hidden agreement" in their underlying confidence distributions. These findings suggest that high disagreement rates under T4 may partly reflect surface-level performance rather than meaningful perspective diversity. Without any downstream quality evaluation — such as human judgments of decision quality, coverage of risk dimensions, or reasoning richness — the claim that Devil's Advocate "works" cannot be substantiated.
3. The experimental design does not model real multi-agent deliberation. The context-isolated architecture, where agents never observe each other's outputs, eliminates the core mechanism by which sycophancy manifests in interactive multi-agent systems: social influence and iterative convergence. The paper is therefore not studying multi-agent disagreement dynamics but rather the variance in parallel independent sampling under different prompts. This is a fundamental mismatch between the paper's framing and its actual experimental setup.
4. The debate paradigm is applied uncritically. Prior work (e.g., Wynn et al., 2025, cited by the authors themselves) demonstrates that multi-agent debate can decrease accuracy. The conditions under which debate is beneficial versus harmful are not discussed, and the paper does not justify why the strategic decision domain — which lacks verifiable ground truth — is an appropriate testbed for evaluating disagreement-inducing techniques.

**Requested Changes:**

1. Justify the value of disagreement. The paper must provide a theoretically grounded argument for why disagreement is beneficial in the specific context of LLM multi-agent systems, distinguishing this from the human group decision-making literature it currently relies on. The relationship between disagreement rate and decision quality must be addressed directly.

2. Add outcome quality evaluation. Disagreement rate alone is insufficient. The authors should include at minimum one form of downstream quality assessment — e.g., expert human evaluation of decision reasoning quality, coverage of alternatives, or identification of overlooked risks — to establish that induced disagreement produces substantively better deliberation rather than performative divergence.

3. Revise claims to match the experimental design. The context-isolated architecture studies parallel independent sampling, not multi-agent deliberation. Either (a) the paper should be reframed accordingly with appropriate caveats, or (b) interactive deliberation conditions should be added to validate findings in a genuine multi-agent setting.

4. Address the inauthentic dissent problem more rigorously. The finding that ~5% of T4 agents argue against their own highest-confidence option, and that ~9% of T4 disagreeing teams have hidden agreement, seriously complicates the paper's recommendation to use Devil's Advocate. The authors should discuss when this is and is not a problem, and whether the technique can be modified to reduce inauthentic dissent.

---

### Review · Reviewer_6SUf · 2026-03-03

**Summary Of Contributions:**

This paper is an empirical work on disagreement-induction via prompting techniques in subjective strategic decision making with multi-agent LLM systems. The primary contribution is systematic comparison on a set of 20 business scenarios. The authors show that the “Devil’s Advocate” prompting technique is superior in inducing disagreement, due to the explicit behavioral assignment. The authors focus strictly on measuring disagreement, while offering some discussion for practical usage.

**Audience:**

No

**Audience Explanation:**

I believe that interest in the TMLR community would be limited. The scope of this paper is narrowly focused on inducing disagreement as a primary outcome metric. The paper does not consider whether the downstream effects of this disagreement is beneficial. Without this, the broader impact of these finding on multi-agent system design remains unclear.

**Claims And Evidence:**

No

**Claims Explanation:**

The main claim of this paper is purely directed towards inducing disagreement. The authors sufficiently report evidence in aggregate (Section 4.1), disagreement/agreement splits and coalition (Sections 4.3 and 4.5), uniqueness of choice (section 4.4), and confidence (section 4.9), among other secondary metrics. I commend the authors for the thorough breakdown of results.

However, the authors provide 480 total experimental runs using a single LLM. Without additional LLMs it is not possible to make a claim that this generalizes (although I do suspect these results do generalize). I am also concerned about the imbalance in sample size for non devil’s advocate methods in table 3 and figure 1 for example. Due to the much lower sample size relative to T4, power to detect significant effects is degraded in comparison to T4, with a comparable n, I believe that it is possible that these effects are statistically significant when comparing to the baseline (i.e. the baseline CI would be narrower as well as T2, T3, and T6). This would change the main claim of this paper considerably.

**Requested Changes:**

1. Standardize (or just increase) the sample size between different methods for fair(er) comparison. The current lack of statistical significance does not imply that other methods have the same performance as the baseline, rather that there is not sufficient evidence to make a conclusion to the alternative. I suspect that this is because the study is underpowered.

2. Utilize a second LLM for experiments. Without sufficient evidence showing generalization to other LLMs, then the scope of this paper is limited to a single model, specifically Sonnet 3.5. Otherwise, the claims should be framed as model specific.

3. Include an experiment that quantifies downstream impact of disagreement. Without this, it is not possible to assess whether there is any utility in using devil’s advocate for this scenario, which limits the impact of this paper. Even if disagreement is achieved in more instances, it is not empirically supported that this disagreement mechanism produces higher quality decisions.

---

### Review · Reviewer_WJ2F · 2026-03-11

**Summary Of Contributions:**

This paper studies how to induce disagreement in multi-agent LLM decision-making systems. the paper conducts a systematic empirical comparison of prompting strategies designed to encourage disagreement based on a simulated executive team. The main finding is that Devil’s Advocate assignment dramatically increases disagreement.

**Strengthes:**
1. The study conducts a large-scale empirical analysis: 480 team decisions and 1,920 individual responses across 20 realistic business scenarios.

2. The identification of the "Devil's Advocate" efficacy is a main finding.


**Weaknesses:**
1. The study uses a fixed team structure (CEO, CFO, CMO, COO), which may bias coalition patterns and limit the generality of the findings.

2. While the paper measures disagreement rates, it does not clearly evaluate whether increased disagreement actually leads to better decisions or improved outcomes.

3. Because the scenarios have no objectively correct answers, the paper measures disagreement as the primary metric. It remains unclear if this disagreement actually leads to "better" business outcomes or just longer conversations.

**Audience:**

Yes

**Audience Explanation:**

As the use of LLM-based agent teams becomes increasingly common in areas such as collaborative reasoning, agent planning, and AI-assisted decision-making, understanding how to avoid premature consensus and encourage diverse viewpoints is an important practical challenge.

**Broader Impact Concerns:**

As multi-agent systems become "decision-support" tools, the ability to induce disagreement could be used to nudging human users toward specific outcomes by "staging" a debate that makes a pre-determined conclusion seem like the hard-won result of a complex deliberation.

The paper does not fully address the "black box" nature of why an agent chooses certain arguments when forced to dissent.

**Claims And Evidence:**

No

**Claims Explanation:**

while the experimental evidence supports the specific claim about disagreement rates, it is less convincing with respect to the broader implications for improving multi-agent deliberation or decision quality.

**Requested Changes:**

1. The current evaluation primarily measures disagreement rates and related behavioral patterns. However, it remains unclear whether increased disagreement actually improves deliberation quality or leads to better decisions. The authors should include additional evaluation metrics assessing the quality or robustness of team decisions.

2. The authors should either temper the generality of the claims or provide additional experiments demonstrating that the findings generalize to other tasks or team compositions

3. The authors should evaluate the effect of disagreement on decision quality.

---

### Decision · Action_Editor_fbnq · 2026-05-18

**Recommendation:** Reject

**Audience:**

Yes

**Audience Explanation:**

The topic of inducing disagreement in multi-agent LLMs is interesting to TMLR's audience, but the interest is limited considering the scope of this paper is narrowly focused on inducing disagreement as a primary outcome metric without evaluating downstream effects.

**Claims And Evidence:**

No

**Claims Explanation:**

The paper studies disagreement-induction via prompting techniques in subjective strategic decision making with multi-agent LLM systems.  All reviewers shared concerns about lack of experiments to support the claims, which is very general and requires fair comparison,  experiments on additional LLMs, and quantification of downstream impacts. Unfortunately, the authors did not provides response or revision to address the concerns.

**Resubmission Of Major Revision:**

The authors may consider submitting a major revision at a later time.